# Elucidation of Anti-Hypertensive Mechanism by a Novel *Lactobacillus rhamnosus* AC1 Fermented Soymilk in the Deoxycorticosterone Acetate-Salt Hypertensive Rats

**DOI:** 10.3390/nu14153174

**Published:** 2022-08-02

**Authors:** Haicui Wu, Lilong Jiang, Tim-Fat Shum, Jiachi Chiou

**Affiliations:** 1Shenzhen Key Lab for Food Biological Safety Control, Hong Kong PolyU Shen Zhen Research Institute, Shenzhen 518000, China; wu.hc.wu@connect.polyu.hk; 2Department of Applied Biology and Chemical Technology, The Hong Kong Polytechnic University, Hong Kong, China; lilong.jiang@polyu.edu.hk (L.J.); tim-fat-perry.shum@connect.polyu.hk (T.-F.S.); 3Research Institute for Future Food, The Hong Kong Polytechnic University, Hong Kong, China

**Keywords:** *L. rhamnosus* AC1, fermented soymilk, hypertension, gut microbiota

## Abstract

Dietary intake of fermented soymilk is associated with hypotensive effects, but the mechanisms involved have not been fully elucidated. We investigated the anti-hypertensive effects of soymilk fermented by *L. rhamnosus* AC1 on DOCA-salt hypertension from the point of view of oxidative stress, inflammatory response and alteration of the gut microbiome. The antioxidant assays in vitro indicated the ethanol extract (EE) of *L. rhamnosus* AC1 fermented soymilk showed better antioxidative effects than the water extract (WE). Those extracts displayed a hypotensive effect using a tail-cuff approach to measuring blood pressure and improved nitric oxide (NO), angiotensin II (Ang II), tumor necrosis factor-α (TNF-α) and interleukin factor-6 (IL-6) on DOCA-salt hypertensive rats. Furthermore, cardiac and renal fibrosis were attenuated by those extracts. The gut microbiota analysis revealed that they significantly reduced the abundance of phylum Proteobacteria, its family Enterobacteriaceae and genus *Escherichia-Shigella*. Moreover, metabolomic profiling revealed several potential gut microbiota-related metabolites which appeared to involve in the development and recovery of hypertension. In conclusion, fermented soymilk is a promising nutritional intervention strategy to improve hypertension via reducing inflammation and reverting dysbiotic microbiota.

## 1. Introduction

As a worldwide health problem, hypertension is a risk factor for cardiovascular, cerebrovascular and renal disease [1]. Secondary hypertension accounts for a small portion of clinical hypertensive cases, suggesting a clear single factor could cause high blood pressure. Approximate 95% of hypertensive cases are identified to be essential hypertension. The physiological mechanisms involved in the development of essential hypertension include cardiac output and peripheral resistance, renin-angiotensin-aldosterone system (RAAS), autonomic nervous system and other factors [2]. DOCA-salt-induced hypertension has been used to describe the natural history of malignant hypertension and the biochemical and hormonal characteristics of each stage of the disease [3]. Administration of DOCA and sodium chloride to rats also provides a reliable rat model of oxidative and inflammatory stress in the cardiovascular system [4].

The approaches to hypertension treatment include changing lifestyle adjustments, such as altering diet habitats [5], increasing regular physical exercise [6], reducing stress [7], and using medications. Soymilk, one of the favorite drinks in Asia, contains abundant proteins and calories but no cholesterol or lactose. Numerous studies show there is an increase in quantity, availability, digestibility, and assimilability of nutrients in food fermented by *Lactobacillus* spp., a common species of lactic acid bacteria (LAB) in the gut [8]. It is reported that *Lactobacillus* fermented soymilk is able to inhibit ACE activity and promote the production of NO in the endothelial cells, hence harboring the potential to regulate blood pressure [9].

Gut microbiota as an “essential organ” in humans harbor approximately 150 times more genes than those in the entire human genome [10,11]. Those gut microorganisms play crucial roles in human health and diseases, including obesity [12], colorectal cancer [13], liver cirrhosis [14], arthritis [15], and type 2 diabetes [16]. Recently, it is indicated that hypertension is linked to gut microbiota in several hypertensive animal models, including spontaneously hypertensive rat (SHR), salt-sensitive hypertensive rat and chronic angiotensin II infusion rat [17] though the mechanisms involved were yet to be elucidated.

Our previous study indicates that *L. rhamnosus* AC1 exhibited superior performance in soymilk fermentation, especially in converting glycoside forms of isoflavone into aglycone forms, showing its potential as a starter and probiotic culture [18]. We also have observed that DOCA-salt hypertension is associated with gut microbiota dysbiosis [19]. In this study, it is hypothesized that the soymilk fermented by the novel *L. rhamnosus* AC1 improves hypertension via altering multiple routes including oxidative stress, inflammatory response and the gut microbiome. In this study, we mainly aim to evaluate the anti-hypertensive effects of this fermented soymilk on DOCA-salt hypertensive rats from the point of view of those aspects.

## 2. Materials and Methods

### 2.1. Bacteria Culture

*L. rhamnosus* AC1 was isolated from an infant fecal sample, cultured in Lactobacilli de Man, Rogosa, and Sharpe (MRS) broth and stored with 20% glycerol at −80 °C.

### 2.2. Preparation of Fermented Soymilk with L. rhamnosus AC1 and Its Extract

Twenty grams of soybeans were soaked in distilled water for 8 h, followed by blending with 1 L of distilled water to make soymilk. It was fermented by 3 × 10^6^ CFU/mL of *L. rhamnosus* AC1 and then dried to a powder with a freeze drier. A total of 30 g of fermented soymilk powder was added with 150 mL of distilled water and incubated at 37 °C for 1 h. The supernatant was then taken by centrifuge and dried to obtain the water extract. Then, 30 g of fermented soymilk powder was added with 150 mL of 100% ethanol and incubated at 37 °C for 1 h. The supernatant was then taken by centrifuge and dried to obtain ethanol extract.

### 2.3. Measurement of Antioxidative Abilities In Vitro

Free radical scavenging activity was determined by adding 100 μL of each sample to 500 μL of 0.184 mM 2,2-Diphenyl-1-picrylhydrazyl (DPPH) in 95% ethanol. The absorbance of the resulting solution was measured at 517 nm. Reduction ability was measured by adding 100 μL of each sample to 100 μL of 1% K_3_Fe(CN)_6_ and 100 μL of 0.2 M phosphate buffer. A total of 500 μL of 10% TCA was added to this mixture and centrifuged. The supernatant was collected to mix with distilled water and 0.1% FeCl_3_ at a ratio of 1:1:1 and the absorbance of the resulting solution was measured at 700 nm. The ferrous ion-chelating activity was measured by adding 200 μL of each sample to 740 μL of methanol and mixed thoroughly. A total of 20 μL of 2 mM FeCl_2_ and 40 μL of 5 mM ferrozine were added to the mixture and incubated at room temperature in dark. The absorbance of the resulting solution was measured at 562 nm.

### 2.4. Animal Experiments

A total of 25 male Sprague Dawley (SD) rats (6-week-old, 180–200 g) purchased from Centralized Animal Facilities (CAF) of Hong Kong Polytechnic University were used in this experiment and housed individually in CAF at a temperature of 21 ± 2 °C and 55% ± 10% relative humidity with a 12:12 h light-dark cycle. They were approved by the Animal Subjects Ethics Committee of The Hong Kong Polytechnic University and were conducted in accord with Hong Kong Government Animal Ordinance Chapter 340. They were supplied with autoclaved corn-cob bedding (Lab Supply, Northlake, TX, USA) and fed ad libitum irradiated diets (LabDiet 5053, Lab Supply, Northlake, TX, USA). Rats were housed for a minimum of two weeks in the facility prior to the experiment and there were 5 rats in each group.

### 2.5. DOCA-Salt Hypertensive Rat Model

Five groups, namely SHAM (normotensive rats), DOCA (DOCA-salt hypertensive rats without treatment), CAP (the DOCA-salt hypertensive rats with captopril administration), WE (the DOCA-salt hypertensive rats with water extract administration) and EE (the DOCA-salt hypertensive rats with ethanol extract administration) were used in this study. Hypertension induction was according to a previous study [19]. This animal experiment lasted for 14 weeks. In the CAP, WE and EE groups, hypertensive rats were given captopril (50 mg/kg), water extract and ethanol extract (2.25 and 0.15 g/kg) by oral gavage daily for the last 5 weeks. The measurements of body mass, food/water intake and blood pressure were recorded in a previous study [19].

### 2.6. Biochemical and Histological Assessment

Blood was collected by cardiac puncture following the sacrifice of the rats by CO_2_ inhalation. Plasma was obtained by collecting blood in heparinized syringes containing 5% heparin and stored at −80 °C. Serum was separated from the blood by centrifugation and stored at −80 °C for metabolomic analysis. The aorta, heart, and kidney tissues were harvested, weighed and stored at −80 °C. NO, Ang II, IL-6 and TNF-α were detected by ELISA kits according to a previous study [19].

Tissues were fixed in 10% neutral buffered formalin, embedded in paraffin, and sectioned to 5 μm of thickness. The percentage of renal tubulointerstitial damage, renal/heart perivascular fibrosis and vascular elastin distribution was analyzed according to previous work [19].

### 2.7. Analysis of Gut Microbiota

Stool samples were collected sterilely from the rectum of rats before sacrifice and stored at −80 °C. DNA was extracted from each sample using the TIANamp Stool DNA kit (Tiangen, Beijing, China) and the V3-V4 region of the bacterial 16S rRNA was sequenced by Illumina PE250/PE300 sequencer (300–500 bp paired-end reads) at Majorbio BioPharm Tech. Co. (Shanghai, China). Raw data were filtered, and primers were trimmed. Results of the sequences were analyzed using QIIME 2 and figures were generated using i-sanger platform. PICRUSt 2 was used to predict molecular functions of microbiota from the databases, KEGG and MetaCyc.36

### 2.8. Metabolomic Profiling of Serum Samples

Serum (50 μL) was precipitated with 200 μL of methanol and vortex-mixed for 2 min. The sample was subsequently centrifuged at 13,000× *g* for 15 min at 4 °C. The supernatant was then dried using a vacuum concentrator (Labconco, Kansas City, MO, USA) and the residue was further reconstituted using 100 μL of 50% methanol containing 1 μg/mL 4-chlorophenylalanine (4-Cl-Phe) as the internal standard (IS). After centrifugation at 13,000 rpm at 4 °C for 15 min, the supernatant was subjected to metabolomics profiling by Thermo Scientific Orbitrap MS coupled with LC (Thermo Fisher Scientific, Austin, TX, USA).

Liquid chromatographic separations were achieved on a UPLC HSS T3 column (2.1 × 100 mm, 1.8 μm, Waters). The mobile phase consisted of 0.1% formic acid-water (*v*/*v*; A) and 0.1% formic acid-acetonitrile (*v*/*v*; B) at the flow rate of 0.3 mL/min with a gradient as follows: 0 to 1 min, 2% B; 1 to 20 min, 2% to 100% B; 20 to 22 min, 100% B; 22 to 22.1 min, 100% to 2% B; 22.1 to 25 min, 2% B. The operation parameters of the mass spectrometer were set as follows: Ion Source Type: ESI; Positive Ion (V): 3500; Negative Ion (V): 2500; Ion Transfer Tube Temp: 350 °C; Vaporizer Temp: 350 °C; Orbitrap resolution: 60,000.

### 2.9. Data Analysis

All values represent the means and standard error of four to six independent experiments. Data were then compared using One Way ANOVA with Duncan’s multiple range analysis in SPSS statistical analysis software (IBM Software, Armonk, NY, USA).

All results were considered statistically significant at *p* < 0.05. Partial least squares–discrimination analysis (PLS-DA) was conducted to identify the discrimination of variables. Differential metabolites were defined as those with variable importance in the projection (VIP) > 1.0, fold change (FC) > 1.2 or <0.8, and *p* values less than 0.05.

## 3. Results

### 3.1. Antioxidative Effects of L. rhamnosus AC1-Fermented Soymilk In Vitro

LAB strains are able to convert the glycoside form of isoflavone into aglycone form in soymilk, the latter exhibit higher antioxidative activities. We evaluated the antioxidative activities of water extract and ethanol extract of *L. rhamnosus* AC1-fermented soymilk by different approaches. The effects of those extracts on DPPH scavenging ability, ferrous ion chelating ability and total reducing activity were shown in Figure 1. Ascorbic acid (ASA, 0.1 mg/mL) and EDTA (0.1 mg/mL) were used as the positive controls and exhibited good performance. Lower dosages (0.1 and 1 mg/mL) of water extract showed higher DPPH radical scavenging activity than those in ethanol extract while higher dosages (5, 10 and 20 mg/mL) of ethanol extract exhibited higher scavenging activity when compared to the same dosages of water extract (Figure 1A). Water and ethanol extract displayed ferrous ion chelating ability in a dose-dependent manner. The chelating activity of ethanol extract was much stronger than that of water extract at the same dosage (Figure 1B). Different from the results of DPPH radical scavenging ability and ferrous chelating activity, the ethanol extract of *L. rhamnosus* AC1-fermented soymilk did not show higher reducing ability in comparison to water extract at high doses (5, 10, 20 mg/mL), though the reducing ability in lower dosages of ethanol extract (0.1 and 1 mg/mL) increased relatively, but not significantly (Figure 1C). In summary, it indicated that the ethanol extract of *L. rhamnosus* AC1-fermented soymilk had overall better effects than the water extract on antioxidative activities in vitro.

### 3.2. Improvement Effects of L. rhamnosus AC1-Fermented Soymilk Extract on Hypertensive Rats

The timeline of the animal experiment was shown in Figure 2A. It started from week 0 to week 14, and all rats were sacrificed at week 15. DOCA were subcutaneously injected into rats at week 1. Their blood pressure was measured biweekly while their body mass, food and water intake were recorded every week. When high blood pressure was successfully induced in the rats at week 9, they were treated with water extract, ethanol extract or captopril, which were sorted as WE, EE and CAP groups (Figure 2A). The average body mass of rats in SHAM increased from 276.16 g to 499.60 g (80.90%) from week 1 to week 14. Nevertheless, the increases in body mass in DOCA, CAP, WE and EE groups were 60.22%, 60.60%, 65.28% and 68.59%, respectively, which were less than that in SHAM (Appendix A
Figure A1). Food intake in SHAM was relatively less than that in the other four groups since week 9 (Appendix A
Figure A1). Water intake in the hypertensive rats was much higher than that in SHAM due to salt supplementation in the drinking water, though the value decreased in CAP since week 10 (Appendix A
Figure A1).

The systolic blood pressure of the SHAM group ranged from 120 to 140 mmHg. In contrast, the systolic BP increased to more than 150 mmHg from week 4 in the other four groups, though it dropped to 140 mmHg in the WE group at week 6 (Figure 2B and Table 1). The blood pressure was approximately 15.30% lower in the CAP group from week 12 to week 14 in comparison to that in the DOCA group (Figure 2B and Table 1). The systolic BP in the WE and EE groups was 5.66% and 9.52%, respectively, lower than that in the DOCA group (Figure 2B and Table 1). Overall, the ethanol extract treatment exhibited higher blood pressure lowering effect than the water extract.

### 3.3. Improvement Effects of L. rhamnosus AC1-Fermented Soymilk Extract on Tissue Hypertrophy and Fibrosis

DOCA, combined with salt loading in the diet of rats, is able to induce hypertension with cardiovascular remodeling characteristics, including hypertrophy and fibrosis [4]. In this study, we observed that the kidneys of hypertensive rats in DOCA, CAP, WE and EE were larger than those in SHAM (data not shown). However, the ratios of kidney weight to body weight (K/B) in CAP, WE and EE groups were lower than those in the DOCA group though the difference is not significant (Figure 2C). As DOCA-salt-induced hypertensive rats also showed heart hypertrophy, the ratio of heart weight to body weight (H/B) was calculated. As expected, the H/B ratio of rats in SHAM was significantly lower than in the other four groups (Figure 2D). The alleviation effect for heart hypertrophy in the hypertensive rats with or without treatments did not exhibit much difference though the H/B ratio of rats in the EE group was lower than that in the DOCA group (Figure 2D). Altogether, the results showed that the *L. rhamnosus* AC1-fermented soymilk extract, especially the ethanol extract, relatively alleviated kidney and heart hypertrophy in DOCA-salt hypertensive rats.

In addition, trichrome staining showed that the level of renal perivascular fibrosis was also improved to a degree by ethanol extract treatment but not by water extract treatment (Figure 3A,B). However, the water/ethanol extract did not exhibit perivascular fibrosis in the heart (Figure 3C,D). Using a modified elastic staining approach to evaluate the elastin fibrosis in the aorta, the results indicated that both water/ethanol extract could ameliorate the fibrosis level, in which the ethanol extract exhibited a better improvement effect (Figure 3E,F). Therefore, all the histological staining results showed that ethanol extract of *L. rhamnosus* AC1-fermented soymilk displayed a better protective effect on tissue fibrosis in DOCA-salt hypertensive rats.

### 3.4. Regulatory Effects of L. rhamnosus AC1-Fermented Soymilk Extract on DOCA-Salt Hypertensive Rats

During cardiovascular remodeling and other chronic pathophysiological stress states, oxidative stress and inflammation are intricately combined [20]. Administration of DOCA and sodium chloride to rats provides a reliable model of hypertension, oxidative and inflammatory stress [4]. Several biochemical changes involved in the pathways of RAAS, the most important pathway in regulating blood pressure, oxidative and inflammatory stress were evaluated in this study. Firstly, NO content was determined by measuring the content of nitrate and nitrite in the plasma. The NO content of the DOCA group was higher than SHAM rats while water/ethanol extract administration significantly improved NO content (Figure 4A). As expected, angiotensin II level was much higher in the DOCA group than that in the SHAM group. It is of interest that the content of angiotensin II in the other two groups, WE and EE, was notably decreased while ethanol extract administration exhibited better performance (Figure 4B).

Two biochemical parameters involved in inflammatory stress, TNF-α and IL-6, were also assessed. Compared to the level of TNF-α in SHAM, all the rats in the other groups exhibited a significant increase in TNF-α. Relatively decreased TNF-α content was observed in rats treated with water/ethanol extract despite the difference was not statistically different (Figure 4C). IL-6 was also increased in hypertensive rats of DOCA, WE and EE groups, yet ethanol extract administration displayed a significant reverting effect to a level similar to the SHAM group (Figure 4D).

All the assessments of biochemical changes demonstrated that both water and ethanol extract administration exhibited a relatively protective effect on the increasing blood pressure of hypertensive rats. We also observed ethanol extract treatment was more effective than water extract on several biochemical parameters involved in RAAS and inflammatory stress.

### 3.5. Effects of L. rhamnosus AC1-Fermented Soymilk Extract on Alteration of Gut Microbiome

The α-diversity, shown as the Sobs index to measure the diversity within individuals, did not exhibit significant changes among all groups, yet their microbial compositions differed (Appendix A
Figure A2). It has been shown by multiple studies that the ratio of Firmicutes to Bacteroidetes (F/B ratio) is correlated with dysbiosis of gut bacteria for many diseases [12,21,22]. A similar observation was also reported in the spontaneously hypertensive rat [23]. In this DOCA-salt hypertensive rat model, the F/B ratio increased as expected in comparison with control rats in SHAM even though the difference is not statistically significant. By contrast, the F/B ratio of hypertensive rats treated with water/ethanol extract slightly decreased (data was not shown). All the rats were dominated by the phyla, Bacteroidetes, Firmicutes, Proteobacteria, and Cyanobacteria at week 0 and week 14 (Figure 5A and Appendix A
Figure A3). In the DOCA group, the percentage of Proteobacteria, a major phylum of gram-negative bacteria and containing a wide variety of pathogens, was the highest in comparison to the rats of the other four groups at week 14 while no differences were found at week 0 in all the groups (Figure 5A and Appendix A
Figure A3). It is interesting that ethanol extract and water extract treatment in DOCA-salt hypertensive rats attenuated the abundance of Proteobacteria in the rat gut microbiota (Figure 5A). At the family level, all the hypertensive rats with or without treatments exhibited less Bacteroidales_S24-7 group, compared to those in SHAM at week 14 (Figure 5B). However, no differences in its abundance were found in all the groups at week 0 (Appendix A
Figure A3).

At the phylum level, the abundance of Proteobacteria was much lower in SHAM (0.6%) compared to that in the other three groups (7.43% in DOCA; 5.08% in WE; 2.327% in EE) (Figure 5B and Figure 6). To further study the bacteria belonging to this phylum, we found that the abundance of class Betaproteobacteria in the SHAM group was much higher than that of Gammaproteobacteria in the phylum Proteobacteria. In contrast to the SHAM group, the abundance of Gammaproteobacteria was much higher than that of Betaproteobacteria in the other 3 groups though WE and EE treatments relatively rebalanced the percentage of Betaproteobacteria in the Proteobacteria phylum (Figure 6). Phylogenetic distribution of gut microbiota revealed that almost no enriched bacteria were found in all the groups though one norank genus belonging to the family Erysipelotrichaceae was abundant in the DOCA group only at week 0 (Appendix A
Figure A4). Notably, the rats in each group were enriched by specific microbial flora at week 14. For example, the order Bacillales, class Gammaproteobacteria, genus *Blautia*, and family Prevotellaceae were most abundant in SHAM, DOCA, WE, and EE groups, respectively (Figure 7). In humans, *Klebsiella*, which belong to the family Enterobacteriaceae, overgrew in pre-hypertensive and hypertensive populations [24]. In our hypertensive rat model, the family Enterobacteriaceae was also proved to be abundant in DOCA-salt hypertensive rats and could be improved by water/ethanol extract treatments. Another abundant genus in pre-hypertensive and hypertensive human beings was *Prevotella*, belonging to the family Prevotellaceae [24]. Nevertheless, the abundance of this family was the highest in EE in comparison to that in other groups (Figure 7). Overall, the results indicated that DOCA-salt hypertensive rats exhibited altered gut microbiota compared to those in normotensive rats in SHAM while our treatments, water and ethanol extract, were able to relatively rebalance the dysbiotic gut flora.

To further study the protective effect of *L. rhamnosus* AC1-fermented soymilk on changes in gut microbiota, their functions of gut microbiota were predicted using PICRUSt2. Tryptophan and glutamic acid are two precursors of neurotransmitters, which could directly impact the central and peripheral nervous systems. Arginine is a precursor for the synthesis of nitric oxide (NO), making it important in the regulation of blood pressure. Phenethylamine functions as a monoaminergic neuromodulator, and to a lesser extent, a neurotransmitter in the human central nervous system. The changes in the abundance of bacteria involved in the metabolism of tryptophan, glutamate, arginine and phenylethylamine were significantly different in SHAM and DOCA groups although the amount of those bacteria shown in this study was quite little. Moreover, water/ethanol extract treatment could relatively, but not significantly, rebalance their abundance (Appendix A
Figure A5). The above results suggested that water/ethanol extract might revert some of the gut bacteria involved in crucial pathways which are associated with the regulation of blood pressure in DOCA-salt hypertensive rats.

### 3.6. L. rhamnosus AC1-Fermented Soymilk Extract Alters the Levels of Potential Gut Microbiota-Associated Metabolites in DOCA-Salt-Induced Hypertension

To assess metabolic profiles in response to the gut microbiota altered by *L. rhamnosus* AC1-fermented soymilk extract, untargeted metabolomic profiling was generated on serum samples by orbitrap LC-MS. After peak picking and normalization, a total of 4982 positive ions and 4994 negative ions were detected in the serum. Distinct clustering of metabolites was apparent among SHAM, DOCA-treated, and extracts-treated DOCA groups by PLS-DA (Figure 8A). By using the comparison between the extraction-treated group vs DOCA group and DOCA group vs SHAM group, we found that extract treatment partially regulated the metabolites altered upon DOCA induction, reversing 13 of the DOCA-induced metabolite changes by water extract treatment (Figure 8B), and 14 of the DOCA-induced metabolite changes by ethanol extract treatment (Figure 8D). Furthermore, we analyzed the correlations between 5 phylum and 14 family bacteria, and these differential metabolites. With the treatment of water extract (Figure 8C), 5 metabolites such as L-pipecolic acid was positively correlated with the family Moraxellaceae, whereas negatively correlated with phylum Tenericutes (*p* < 0.05). 7-keto-25-hydroxycholesterol was positively correlated with the family Enterobacteriaceae, whereas negatively correlated with phylum Tenericutes (*p* < 0.05). 7α-hydroxy-3-oxo-4-cholestenoic acid was positively related to 3 gut microbes, including phylum Enterobacteriaceae, phylum Proteobacteria and family Moraxellaceae. Alpha-*N*-phenylacetyl-L-glutamine was positively related to the family Prevotellaceae. DOPC was negatively correlated with the family Alcaligenaceae and Lactobacillaceae. On the other side, 13(S)-HODE, 13-HDoHE and Pinolenic acid were positively correlated with the family Lachnospiraceae and phylum Firmicutes with the treatment of ethanol extract (*p* < 0.05) (Figure 8E). On the contrary, 13(S)-HODE and Pinolenic Acid were negatively correlated with the family Peptostreptococcaceae (*p* < 0.05). Family Alcaligenaceae was positively correlated with alpha-*N*-phenylacetyl-L-glutamine (*p* < 0.05) and choline (*p* < 0.01). Family Moraxellaceae and phylum Tenericutes were negatively correlated with Eicosapentaenoic Acid (*p* < 0.01) and L-Pipecolic acid (*p* < 0.05), respectively. On the other hand, the family Christensenellaceae and phylum Proteobacteria were positively correlated with 3-methyl pyruvic acid and ornithine (*p* < 0.05), respectively. Together, these results indicated that some bacteria and their potential derived metabolites were associated with the development and recovery of DOCA-induced hypertension.

## 4. Discussion

Hypertension, a growing global health problem, affects over 1.2 billion individuals and leads to appropriately 40% of cardiovascular mortalities worldwide [25]. It is a multifactorial disease resulting from environmental and genetic factors together. The etiology of the disease in 90% of the cases is not fully clear, which is called essential hypertension [25]. The DOCA-salt hypertension illustrates the crosstalk between the nervous system and hormones which leads to hypertension. Furthermore, it is one of the common models used to study the essentials of hypertension [26]. Cardiovascular remodeling includes hypertension, hypertrophy and fibrosis resulting from chronic pathophysiological progression. Similar to the cardiovascular remodeling characteristic of human volume-overload induced hypertension, DOCA-salt hypertensive rat model induces hypertension with tissue hypertrophy, fibrosis and endothelial dysfunction [27,28,29,30,31,32,33]. Consistently, in our study, DOCA-salt-induced hypertensive SD rats exhibited significant hypertrophy index and fibrosis levels in both the kidney and heart in comparison to those in the SHAM group while the treatments, water/ethanol extract, relatively improved those parameters. Complex biochemical pathways which are associated with inflammatory reactions and the generation of reactive free radicals also cause cardiovascular structural remodeling [4]. Oxidative stress and inflammation intricately combined to elicit a chronic pathophysiological stress state contribute to the pathology and progression of hypertension. The NO content was reduced in DOCA-salt hypertensive rats, but the value was partially restored by water/ethanol extract treatments. Conversely, the SOD activities among all the groups were similar (data not shown). The pathways of NO synthase and SOD in this model are probably different. In addition, the level of TNF-α and IL-6, two inflammatory factors, increased in the hypertensive rats but were attenuated by our treatments. These results indicate that the DOCA-salt hypertensive SD rat model could be a reliable model to study the relationship between oxidative stress, inflammation, and cardiovascular disease.

Although it is not yet clear whether changes in gut microbiota are the causation or consequence of hypertension, nor the mechanisms involved are fully elucidated, increasing evidence suggests that there is a strong relationship between the gut microbiota and hypertension [17,23,24,34,35]. Isoflavones as the main beneficial components in soymilk have two major forms, aglycones and glycosides [36]. Since the structure itself is a limiting factor for absorption in the gastrointestinal tract, the chemical forms of the aglycones are more readily absorbed and more bioavailable than glycoside forms [36]. The glycoside forms could be hydrolyzed into aglycone forms by β-glucosidases of gut microflora to be absorbed in vivo [37]. This knowledge has led to the development of aglycone-enriched products, obtained directly by β-glucosidase treatment or by fermentation with LAB [38]. Our previous study demonstrated that soymilk fermented by *L. rhamnosus* AC1 isolated from infant fecal samples exhibited the highest β-glucosidase activity [18]. Genistein belonging to aglycones in fermented soymilk exhibited a higher antioxidative ability to stimulate the expression of catalase (CAT) or superoxide dismutase (SOD), which might be associated with decreased occurrence of cancer [39]. Fermented soymilk by LAB, in general, significantly enhanced the antimutagenicity of soymilk [40]. It is of interest that *Lactobacillus* fermented soymilk is able to inhibit ACE activity and promote the production of NO in the endothelial cells [9]. *L. rhamnosus* AC1 fermented soymilk in our study, especially EE, showed considerable high antioxidative abilities in vitro. Furthermore, the blood pressure lowering effect of water/ethanol extract of *L. rhamnosus* AC1 fermented soymilk was observed.

A link between the gut microbiome and hypertension has been found since the proof-of-concept study by Li et al., in which study the germ-free mice receiving fecal microbiota transplantation (FMT) from hypertensive patients showed elevated blood pressure [24]. Furthermore, the gut microbiome dysbiosis and the changes in the F/B ratio were reported in several hypertensive animal models [17,24,41]. Our DOCA-salt hypertensive rats also exhibited gut microbiome dysbiosis, which was rebalanced relatively by water/ethanol extract treatments, though the F/B ratio did not change significantly probably due to limited numbers of rats in each group. Acetate- and butyrate-producing bacteria are decreased in SHR while an overgrowth of bacteria such as *Prevotella* and *Klebsiella*, and disease-linked bacteria are found in both pre-hypertensive and hypertensive populations [24]. Changes in the abundance of short-chain fatty acids (SCFAs) producers in the gut microbiota were not observed in those groups, nevertheless, the abundance of some bacteria belonging to the phyla, Proteobacteria, Cyanobacteria and Tenericutes, were able to be relatively rebalanced by water/ethanol extract in hypertensive rats. Our metabolomics data also indicated several microbiota-associated metabolites, including lipid, fatty acid or amino acid, were involved in the development and recovery of hypertension. Further studies are needed to validate the function of these metabolites and gut microbes and their relationships with the onset and progression of hypertension.

The proposed mechanism of administration of *L. rhamnosus* AC1 fermented soymilk to improve hypertension in DOCA-salt hypertensive rats was shown in Figure 9. Aglycone isoflavones were the main form of isoflavones in *L. rhamnosus* AC1 fermented soymilk and exhibited high anti-oxidative ability in vitro. In DOCA-salt hypertensive rats, *L. rhamnosus* AC1 fermented soymilk improved several biochemical parameters in pathways of RAAS, oxidative stress and inflammation, including decreased Ang II, TNF-α and IL-6 levels, increased NO level and SOD activity. Moreover, aglycone isoflavones are mainly produced in the gastrointestinal tract by specific strains and might have an effect on the structural or functional changes of gut microbiota, reshaping the abundance of different gut microbes and/or their derived metabolites. For instance, equol, a downstream metabolite of isoflavones, is produced by gut microbiota. We proposed that the anti-hypertensive effect of *L. rhamnosus* AC1 fermented soymilk is due to the anti-oxidative and anti-inflammatory function of aglycone isoflavones, as well as its effect on the alteration of the gut microbiome. The detailed pathway was not clear and will be further investigated.

A novel strain, *Lactobacillus rhamnosus* AC1, isolated from baby feces exhibited superior performance in soymilk fermentation, especially in converting glycoside forms of isoflavone into aglycone forms. Those nutrients could be further metabolized into downstream active molecules by the enzymes of gut bacteria. This study firstly revealed the anti-hypertensive effects of soymilk fermented by *L. rhamnosus* AC1 on DOCA-salt hypertension from the point of view of the gut microbiome. The purification and absorption of these functional or bioactive ingredients in water and ethanol extracts are interesting topics for future studies, thereby extending the range of applications of soybeans. Undeniably, this study only showed general alterations of gut microbiota and their metabolites in the DOCA-salt hypertensive rodent model, which may not be applicable to humans. Nonetheless, the multidisciplinary approaches of this study demonstrate a good platform for studying the role of probiotics and their fermented products on the selected hypertensive animal model. To further elucidate the function of *L. rhamnosus* AC1 fermented soymilk on hypertension, specific strains or metabolites involved in the development and recovery of hypertension should be investigated in the future.

## 5. Conclusions

Taken together, the fermented soymilk by a novel probiotic strain, *L. rhamnosus* AC1, exhibited good performance on antioxidative ability in vitro, such as high DPPH scavenging and ferrous ion chelating ability. The anti-hypertensive effect was shown in DOCA-salt hypertensive rats by the extract of *L. rhamnosus* AC1 fermented soymilk. The biochemical parameters, including NO, AngII, TNF-α, and IL-6, and tissue fibrosis levels have been improved by water and ethanol extract. In addition, they rebalanced the abundance of phylum Proteobacteria, its family Enterobacteriaceae and genus *Escherichia-Shigella* in the gut microbiota of hypertensive rats. Potential gut microbiota-related metabolites revealed by metabolomic profiling might participate in the development and recovery of hypertension. Thus, this fermented soymilk might be an alternative intervention approach to treat hypertension.

## Figures and Tables

**Figure 1 nutrients-14-03174-f001:**
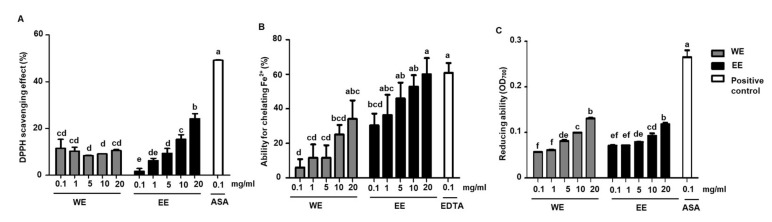
Antioxidant abilities of water extract (WE) and ethanol extract (EE) of *L. rhamnosus* fermented soymilk in vitro. The scavenging effect of DPPH radicals (**A**); The ability for chelating Fe^2+^ (**B**); The reducing activity (**C**). Data presented as means ± SEM (*n* = 2). Bar values with different letters were significantly different by Duncan’s multiple range test (*p* < 0.05). ASA: Ascorbic acid; EDTA: Ethylenediaminetetraacetic acid.

**Figure 2 nutrients-14-03174-f002:**
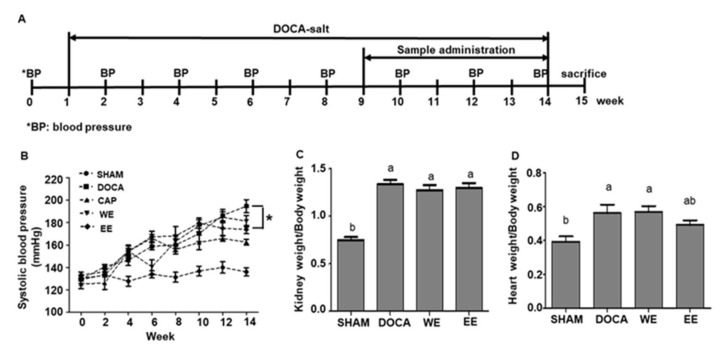
Effects of administration of water extract (WE) and ethanol extract (EE) of *L. rhamnosus* AC1-fermented soymilk on DOCA-salt hypertensive rat. Timeline of animal experiments (**A**). Results of systolic blood pressure (**B**), the ratio of organ weight to body weight in the case of the kidney (**C**) and heart (**D**). Data are presented as means ± SEM (*n* = 5). Bar values bearing different letters were significantly different by Duncan’s multiple range test (*p* < 0.05). *, differences between the selected values were statistically different by Duncan’s multiple range test (*p* < 0.05).

**Figure 3 nutrients-14-03174-f003:**
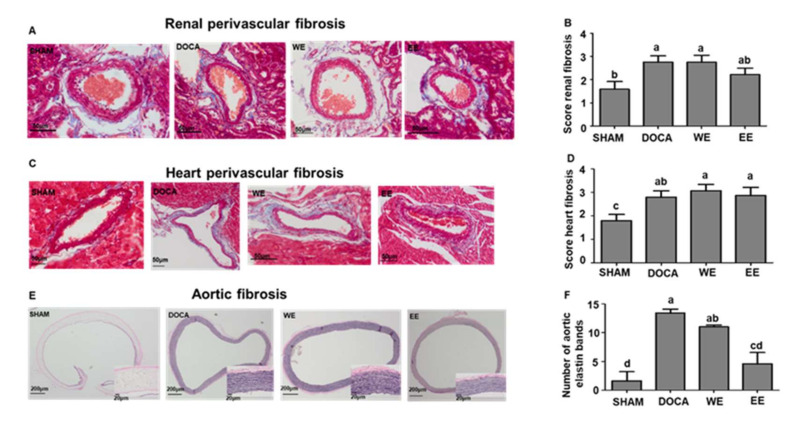
Effects of the water extract and ethanol extract of *L. rhamnosus* AC1-fermented soymilk on tissue fibrosis in DOCA-salt hypertensive rats. Representative figure and quantification of renal perivascular fibrosis (**A**,**B**); Representative figure and quantification of heart perivascular fibrosis (**C**,**D**); Representative figure and quantification of aortic fibrosis (**E**,**F**). Data are presented as means ± SEM (*n* = 6). Bar values bearing different letters were significantly different by Duncan’s multiple range test (*p* < 0.05).

**Figure 4 nutrients-14-03174-f004:**
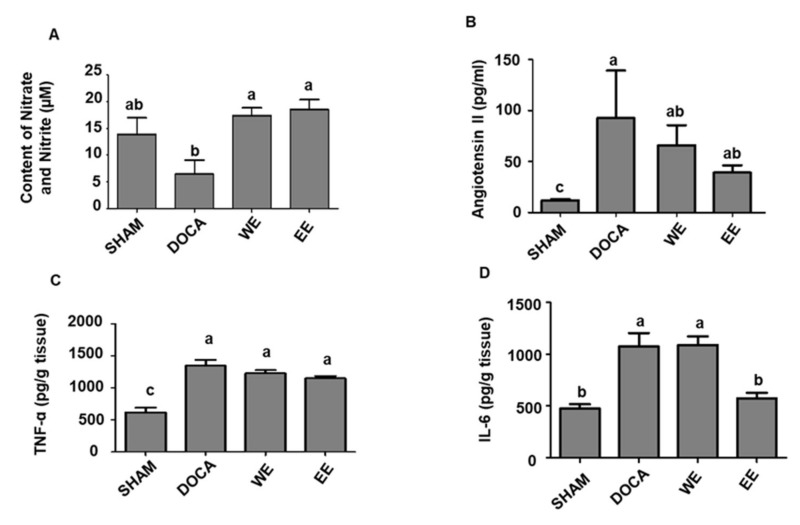
Effects of the water extract and ethanol extract of *L. rhamnosus* AC1-fermented soymilk on the NO content (**A**), levels of angiotensin II (**B**), TNF-α (**C**) and IL-6 (**D**) in the DOCA-salt hypertensive rats. Data are presented as means ± SEM (*n* = 6). Bar values bearing different letters were significantly different by Duncan’s multiple range test (*p* < 0.05).

**Figure 5 nutrients-14-03174-f005:**
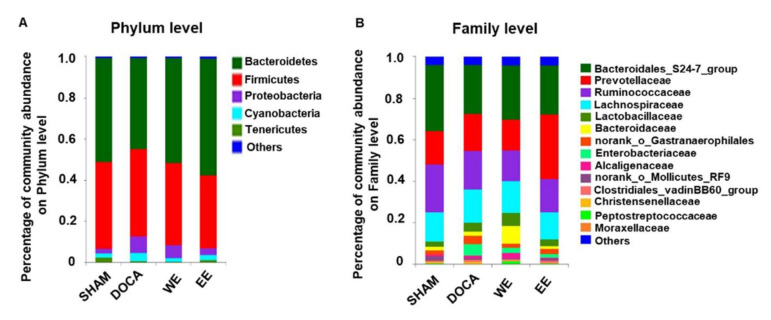
Effect of water extract and ethanol extract of *L. rhamnosus* AC1-fermented soymilk on the microbial composition in the gut flora of DOCA-salt hypertensive rats at week 14. Percentage of community abundance on the phylum (**A**) and family levels (**B**).

**Figure 6 nutrients-14-03174-f006:**
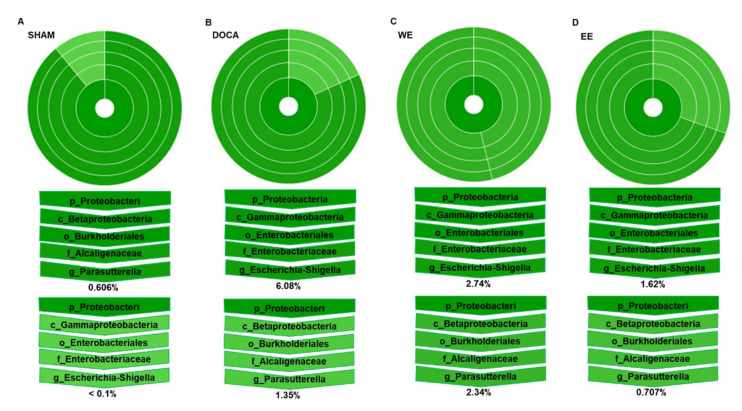
Effect of water extract and ethanol extract of *L. rhamnosus* AC1-fermented soymilk on the microbial composition belonging to phylum Proteobacteria at week 14. Percentage of community abundance in SHAM (**A**), DOCA (**B**), WE (**C**) and EE (**D**).

**Figure 7 nutrients-14-03174-f007:**
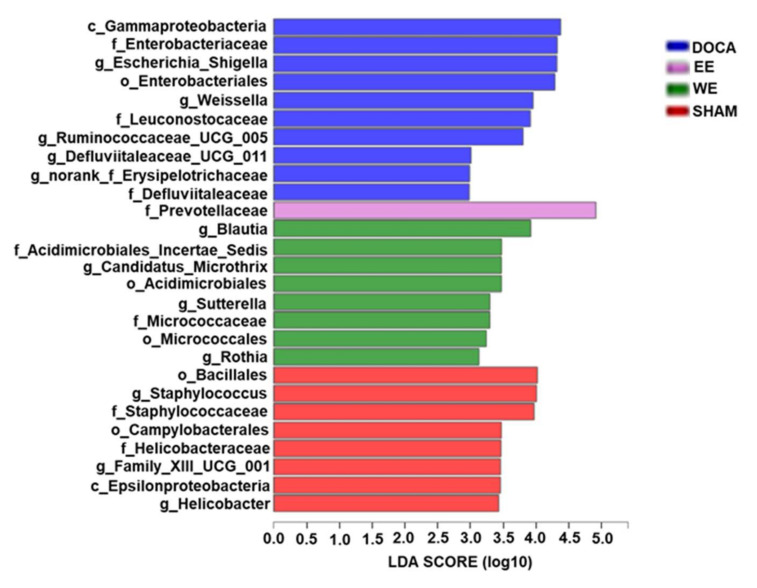
Phylogenetic distribution of gut microbiota of rats in SHAM, DOCA, WE and EE at week 14. Indicator bacteria with LDA scores of 2 or greater in bacterial communities.

**Figure 8 nutrients-14-03174-f008:**
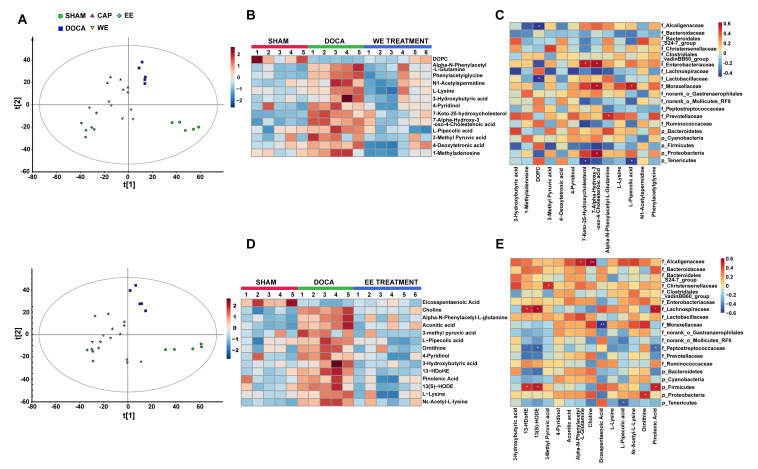
*L. rhamnosus* AC1-fermented soymilk extract alters potential gut microbiota-associated metabolites levels in DOCA-salt-induced hypertension. (**A**) Partial least squares-discrimination analysis (PLS-DA) score plots for discriminating the serum metabolome from Sham, DOCA, CAP, WE-treated and EE-treated groups (Upper: negative mode; Lower: positive mode). (**B**) Heatmap of the differential metabolites that were altered by DOCA and then regulated by WE. (**C**) Heatmap analysis of the spearman correlation of differential metabolites and gut microbiota after WE administration. Red represents positive correlation, and blue indicates negative correlation. (**D**) Heatmap of the differential metabolites that were altered by DOCA and then regulated by EE. (**E**) Heatmap analysis of the spearman correlation of differential metabolites and gut microbiota after EE administration. Red represents positive correlation, and blue indicates negative correlation. *n* = 4–6. * *p* < 0.05, ** *p* < 0.01. WE: water extract. EE: ethanol extract.

**Figure 9 nutrients-14-03174-f009:**
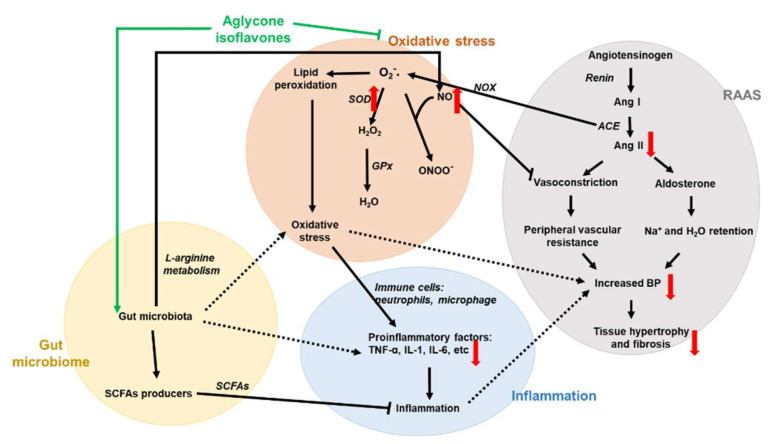
Proposed mechanism of *L. rhamnosus* AC1-fermented soymilk on DOCA-salt hypertensive rat. Four pathways are presented in this study, which are gut microbiota (yellow), oxidative stress (orange), RAAS (gray) and inflammation (blue). All the parameters are shown in the specific pathway. Black solid arrows represent a direct link between two parameters and black dotted arrows donate an indirect link. Red arrows mean the alteration of several biochemical parameters we evaluate. Green arrows indicate the active constituent in our fermented soymilk.

**Table 1 nutrients-14-03174-t001:** Effects of administration of the water extract and ethanol extract of *L. rhamnosus* AC1-fermented soymilk on blood pressure in DOCA-salt hypertensive rats.

Group	Systolic BP (mmHg)							
	Week 0	Week 2	Week 4	Week 6	Week 8	Week 10	Week 12	Week 14
SHAM	129.24 ± 2.55 ^Aa^	133.42 ± 3.14 ^ABa^	128.06 ± 2.59 ^Ba^	138.30 ± 4.11 ^Ba^	131.38 ± 5.62 ^Ba^	139.91 ± 5.22 ^Ba^	140.09 ± 4.32 ^Ca^	136.1 ± 5.06 ^Ca^
DOCA	129.40 ± 3.26 ^Ae^	126.03 ± 4.19 ^Ade^	147.49 ± 5.26 ^Acd^	161.09 ± 6.89 ^Abc^	163.66 ± 6.99 ^Abc^	170.13 ± 8.69 ^Ab^	186.42 ± 8.02 ^Aa^	192.14 ± 9.54 ^Aa^
CAP	125.39 ± 3.59 ^Ab^	140.73 ± 5.29 ^Bb^	156.43 ± 5.69 ^Aa^	166.15 ± 7.89 ^Aa^	155.78 ± 7.96 ^Aa^	162.14 ± 10.20 ^Aa^	164.25 ± 8.65 ^Ba^	162.57 ± 7.23 ^Ba^
WE	130.43 ± 6.52 ^Ac^	133.95 ± 5.23 ^ABc^	155.77 ± 7.85 ^Ac^	147.50 ± 9.46 ^Bb^	166.97 ± 5.87 ^Ab^	178.63 ± 5.99 ^Aa^	190.94 ± 9.84 ^Aa^	181.26 ± 8.32 ^Aa^
EE	133.34 ± 5.41 ^Ad^	136.14 ± 4.69 ^Ad^	159.40 ± 7.23 ^Ac^	163.34 ± 8.26 ^Abc^	164.70 ± 7.46 ^Abc^	178.19 ± 9.56 ^Aa^	175.03 ± 10.21 ^ABa^	173.84 ± 9.65 ^Ba^

Data are presented as means ± SD (*n* = 5). Values with different uppercase and lowercase letters in the same column and in the same row were significantly different by Duncan’s multiple range test (*p* < 0.05).

## Data Availability

Not applicable.

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
