# Peer review of "Elucidation of Anti-Hypertensive Mechanism by a Novel Lactobacillus rhamnosus AC1 Fermented Soymilk in the Deoxycorticosterone Acetate-Salt Hypertensive Rats"

_nutrients, 2022, doi:10.3390/nu14153174_

Round 1

Reviewer 1 Report

The present study shows the application of a novel Lactobacillus rhamnosus fermented soymilk in anti-hypertension effect. The author provides new insight of microbial-metabolomic profiles involved in the development and recovery of the hypertension by intervention of probiotic. The manuscript was well addressed, I would like to suggested minor revision.

1.      Line 22-23 I would like to suggest more specific description other than “rebalanced”.

2.     Figure 5A is not necessary, you may move it to supplementary files or just delete.

3.     The resolution of figures should be improved especially for Figure 6.

4.     α-diversity should be presented in the microbial analysis. 

5.     Line 456 It would be better if you assigned a function rather than “specific”

Author Response

Comment 1: Line 22-23 I would like to suggest more specific description other than “rebalanced”.

Reply: Thank you for the comment.  The word “rebalanced” has been revised to a more specific description “significantly reduced” in lines 23.

Comment 2: Figure 5A is not necessary, you may move it to supplementary files or just delete.

Reply: The Figure 5A has been deleted as suggested. The figure legend and main text have been revised accordingly in lines 292-305.

Comment 3: The resolution of figures should be improved especially for Figure 6.

Reply: Thank you for the suggestion. We have improved the resolution of Figure 6 and Figure 8 as suggested.

Comment 4: α-diversity should be presented in the microbial analysis.

Reply: Thank you for the comment. As the α-diversity of gut microbiota did not exhibit significant change in those groups, the figure has been added in Appendix, Figure A2, as suggested.

Comment 5: Line 456 It would be better if you assigned a function rather than “specific”

Reply: Thank you for the suggestion. We have deleted the word “specific” and added detailed description of a function in lines 498-502.

Reviewer 2 Report

Comments in the attachment.

Author Response

Comment 1: In the 'Results' section, there are single elements of the discussion (references to literature, e.g. 24, 25) that are redundant

Reply: Thank you for the suggestion. We have deleted the redundant discussion and references in the “Results” as suggested in lines 306-308. Also, the sequence of other references has been changed accordingly.

Comment 2: The conclusions from the research should be determined and clearly formulated.

Reply: The conclusion part has been added in the first paragraph of “Discussion” in lines 412-422 as suggested.

Comment 3: At the end of the 'Discussion' section, please indicate the strengths and limitations of the work.

Reply: The strengths and limitations of this work have been added at the end of “Discussion” in lines 506-521 as suggested.

Comment 4: The work requires extensive stylistic i grammatical correction. I would suggest consulting a native speaker for proof editing, especially since the text is of great scientific value but the linguistic aspects make the msc difficult to understand.

Reply: Thank you very much for the suggestion. We have sought the assistance of a native speaker for proof editing to smoothen the content.